# Rethinking Neural Multi-Objective Combinatorial Optimization via Neat Weight Embedding

**Jinbiao Chen[1], Zhiguang Cao[2], Jiahai Wang[1,3,4,*], Yaoxin Wu[5],**
**Hanzhang Qin[6], Zizhen Zhang[1], Yue-Jiao Gong[7]**

[1]School of Computer Science and Engineering, Sun Yat-sen University, P.R. China
[2]School of Computing and Information Systems, Singapore Management University, Singapore
[3]Key Laboratory of Machine Intelligence and Advanced Computing, Ministry of Education,
Sun Yat-sen University, P.R. China
[4]Guangdong Key Laboratory of Big Data Analysis and Processing, Guangzhou, P.R. China
[5]Department of Industrial Engineering & Innovation Sciences, Eindhoven University of Technology
[6]Department of Industrial Systems Engineering & Management, National University of Singapore
[7]School of Computer Science and Engineering, South China University of Technology, P.R. China
`chenjb69@mail2.sysu.edu.cn, zgcao@smu.edu.sg`
`wangjiah@mail.sysu.edu.cn, y.wu2@tue.nl, hzqin@nus.edu.sg`
`zhangzzh7@mail.sysu.edu.cn, gongyuejiao@gmail.com`

## Abstract

Recent decomposition-based neural multi-objective combinatorial optimization (MOCO) methods struggle to achieve desirable performance. Even equipped with complex learning techniques, they often suffer from significant optimality gaps in weight-specific subproblems. To address this challenge, we propose a neat weight embedding method to learn weight-specific representations, which captures weight-instance interaction for the subproblems and was overlooked by most current methods. We demonstrate the potentials of our method in two instantiations. First, we introduce a succinct addition model to learn weight-specific node embeddings, which surpassed most existing neural methods. Second, we design an enhanced conditional attention model to simultaneously learn the weight embedding and node embeddings, which yielded new state-of-the-art performance. Experimental results on classic MOCO problems verified the superiority of our method. Remarkably, our method also exhibits favorable generalization performance across problem sizes, even outperforming the neural method specialized for boosting size generalization.

## 1 Introduction

Multi-objective combinatorial optimization (MOCO) problems (Lust & Teghem, 2010; Zajac & Huber, 2021; Ishibuchi et al., 2015; Türkyılmaz et al., 2020; Liu et al., 2020) have garnered extensive interest within the computational intelligence community due to their widespread applicability across various industries such as logistics, manufacturing, and warehousing. These sectors often require decision makers to consider and accommodate multiple intricate factors concurrently like the costs, fairness, and customer satisfaction. Building upon the foundation of NP-hard single-objective combinatorial optimization (CO), MOCO presents an even greater challenge by involving multiple conflicting objectives that cannot be optimized simultaneously. The primary goal of MOCO is to identify a collection of Pareto optimal solutions, known as the *Pareto set*, simultaneously pursuing both *convergence* (or optimality) and *diversity*.

Due to the NP-hard complexity of MOCO, exact methods (Ehrgott et al., 2016; Bergman et al., 2022) often necessitate exponentially growing computational time as the problem size increases. In response, heuristic methods (Blot et al., 2018) have been adopted to approximate Pareto optimal solutions more efficiently. However, traditional heuristics heavily rely on domain-specific knowledge and require substantial manual tuning for each unique problem. Moreover, they may also involve prolonged solving time due to the intensive inherent iterative search from scratch for each instance.

---

*Corresponding Author.

Recent advancements in deep reinforcement learning have catalyzed the rapid development of *neural MOCO* methods (Li et al., 2021; Zhang et al., 2021; Lin et al., 2022a; Ye et al., 2022; Zhang et al., 2023c; Chen et al., 2023a;b; Wang et al., 2024; Fan et al., 2024; Su et al., 2024; Ye et al., 2025), where deep models are leveraged to autonomously learn promising policies from extensive problem instances. These neural methods are able to mitigate the need of laborious problem-specific design, reduce solving time, and generalize to unseen instances. In this context, the *decomposition* scheme plays a crucial role due to its universality and efficacy (Lin et al., 2024; 2022b; Navon et al., 2021), which is used by most existing neural MOCO methods. Typically, an MOCO problem is decomposed into a series of scalarized subproblems, each associated with a specific *weight* (or preference) vector, and these subproblems are then solved end-to-end using deep models.

To tackle the weight-specific subproblems, neural MOCO methods typically integrate single-objective *neural CO* methods with additional mechanisms. A naive way is to train or fine-tune a separate single-objective model for each decomposed subproblem using transfer learning (Li et al., 2021; Zhang et al., 2021) or meta learning (Zhang et al., 2023c; Chen et al., 2023a), known as the *multi-model* method. However, these methods are impractical due to extensive training or fine-tuning overhead and limited adaptability with predetermined weight vectors. An alternative way is to train only one model, which is used to deal with all subproblems, known as the *single-model* method. For instance, the multi-objective routing attention model (MORAM) (Wang et al., 2024) can only handle predefined weight vectors and only address the multi-objective traveling salesman problem, while the preference-conditioned multi-objective combinatorial optimization (PMOCO) (Lin et al., 2022a) employs a hypernetwork to adapt decoder parameters to any weight vector, offering higher flexibility. Nonetheless, PMOCO struggles with subproblem optimality due to the weight-agnostic encoder and the complexity introduced by the hypernetwork. The recent conditional neural heuristic (CNH) (Fan et al., 2024) realizes a unified model across various sizes, regarded as the state-of-the-art (SOTA) method. However, due to the extra introduction of a complex size-aware decoder, it suffers from the demand of more computational resource and the unscalable problem size embedding for larger sizes.

In short, despite these complicated techniques, neural MOCO methods still exhibit considerable performance gaps. The key to effectively solving these subproblems lies in capturing the *weight-instance interaction*, i.e., simultaneously leveraging both weight and instance (defined by a set of nodes) information. In this sense, we propose a neat method called *weight embedding*, wherein the weight-specific representations are directly learned by the single-objective model to effectively manage the subproblems. Unlike existing ones that feature complex schemes, our method not only achieves new SOTA performance but also does so with remarkable elegance.

Our contributions are summarized as follows. (1) We propose a neat weight embedding method for MOCO. It directly learns weight-specific representations, diverging from previous ones that rely on complex auxiliary techniques. (2) We design two models to instantiate our method. Particularly, we first introduce a succinct weight embedding model that utilizes only addition in its embedding process, and then we present an enhanced weight embedding model with conditional attention. (3) Extensive experimental results on various MOCO problems show that our method not only surpasses the SOTA neural methods in performance but also exhibits superior generalization capabilities across different problem sizes, even outperforming the baseline tailored for generalization with size embedding. Due to its elegance and superiority, our weight embedding is expected to become a fundamental method in the field of neural MOCO, paving the way for the development of more advanced methodologies.

## 2 RELATED WORKS

**Traditional MOCO methods.** Traditional MOCO methods typically fall into two categories: exact and heuristic ones. Exact methods (Ehrgott et al., 2016; Bergman et al., 2022) can deliver accurate Pareto optimal solutions, but often require exponentially increasing computation time. Consequently, heuristic methods, particularly the multi-objective evolutionary algorithms (MOEAs) (Tian et al., 2021; Falcón-Cardona et al., 2021), have become favored alternatives. In this context, dominance-based MOEAs (Deb et al., 2002; Deb & Jain, 2013; Deng et al., 2022) and decomposition-based MOEAs (Zhang & Li, 2007; Qi et al., 2014; Yuan et al., 2016) are recognized as two representative schemes. Furthermore, several MOEAs incorporate specialized local search mechanisms to enhance their efficacy (Jaszkiewicz, 2002; Shi et al., 2020; 2024). Despite extensive research, MOEAs still face hurdles due to the heavy hand-crafted workload and massive solving time.

**Neural CO methods.** Neural CO methods (Garmendia et al., 2024; Zhang et al., 2023a; Mazyavkina et al., 2021; Bengio et al., 2021; Yan et al., 2022) have gained prominence recently, which leverage deep learning models with an encoder-decoder architecture to quickly construct high-quality solutions end-to-end. A pivotal innovation in this field is the attention model (AM) (Kool et al., 2019), which employs the Transformer architecture (Vaswani et al., 2017) to instantiate a new approach in solving CO such as vehicle routing problems. AM has spurred a series of advancements (Kim et al., 2022; Bi et al., 2022; Chen et al., 2022; Zhang et al., 2022; 2023b; Zhou et al., 2023; Grinsztajn et al., 2023; Luo et al., 2023; Chalumeau et al., 2023; Drakulic et al., 2023; Fang et al., 2024; Xiao et al., 2024a;b), including the policy optimization with multiple optima (POMO) (Kwon et al., 2020), which capitalizes on symmetries in the solution space to further narrow the optimality gaps. Due to its competitive performance and versatility, POMO has become a favored base model used in most existing neural MOCO methods.

**Neural MOCO methods.** The neural MOCO methods (as summarized in Appendix A), most of which are based on decomposition, can be broadly categorized into the multi-model and single-model method. The former trains (Li et al., 2021; Zhang et al., 2021) or fine-tunes (Zhang et al., 2023c; Chen et al., 2023a) a separate single-objective model for each decomposed subproblem, while the latter only trains one model for all subproblems (Wang et al., 2024; Lin et al., 2022a). One small part of CNH (Fan et al., 2024) is somewhat technically similar to ours, i.e., inputing the weight vector into the model. However, CNH overall pursues complex techniques such as the size-aware decoder to realize cross-size generalization, which is contrary to our "neat" principle. Different from above decomposition-based neural methods focusing on convergence, another orthogonal line of work focuses on diversity enhancement (Chen et al., 2023b) with increased computational resources. This paper aims to improve the convergence of decomposition-based neural methods by more elegantly utilizing the weight vectors.

## 3 PRELIMINARY

### 3.1 MOCO

An MOCO problem with $M$ objectives can be defined as $\min_{\boldsymbol{x} \in \mathcal{X}} \boldsymbol{f}(\boldsymbol{x}) = (f_1(\boldsymbol{x}), f_2(\boldsymbol{x}), \ldots, f_M(\boldsymbol{x}))$, where $\mathcal{X}$ is a discrete decision space.

**Definition 1 (Dominance).** A solution $\boldsymbol{x}^1 \in \mathcal{X}$ is said to dominate another $\boldsymbol{x}^2 \in \mathcal{X}$, denoted as $\boldsymbol{x}^1 \prec \boldsymbol{x}^2$, if and only if $f_i(\boldsymbol{x}^1) \leq f_i(\boldsymbol{x}^2), \forall i \in \{1, \ldots, M\}$ and $f_j(\boldsymbol{x}^1) < f_j(\boldsymbol{x}^2), \exists j \in \{1, \ldots, M\}$.

**Definition 2 (Pareto optimality).** A solution $\boldsymbol{x}^* \in \mathcal{X}$ is Pareto optimal if it is not dominated by any other solution $\boldsymbol{x}' \in \mathcal{X}$. The set composed of all Pareto optimal solution is called *Pareto set*, i.e., $\mathcal{P} = \{\boldsymbol{x}^* \in \mathcal{X} \mid \nexists \boldsymbol{x}' \in \mathcal{X} : \boldsymbol{x}' \prec \boldsymbol{x}^*\}$, and its image in the objective space is called *Pareto front*, i.e., $\mathcal{F} = \{\boldsymbol{f}(\boldsymbol{x}) \in \mathcal{R}^M \mid \boldsymbol{x} \in \mathcal{P}\}$.

### 3.2 DECOMPOSITION-BASED NEURAL MOCO METHODS

Decomposition (Zhang & Li, 2007) is a mainstream strategy in the neural MOCO field, where an MOCO problem is decomposed into $N$ subproblems by $N$ weight vectors. Each subproblem is a CO problem with a scalarized objective $g(\boldsymbol{x}|\boldsymbol{\lambda})$, where $\boldsymbol{\lambda} \in \mathcal{R}^M$ is a weight vector satisfying $\lambda_m \geq 0$ and $\sum_{m=1}^{M} \lambda_m = 1$. As the simplest representative, the weighted sum (WS) scalarization uses the linear combination of $M$ objectives, i.e., $\min_{\boldsymbol{x} \in \mathcal{X}} g_{\mathrm{ws}}(\boldsymbol{x}|\boldsymbol{\lambda}) = \sum_{m=1}^{M} \lambda_m f_m(\boldsymbol{x})$.

Given $N$ weight vectors, the corresponding scalarized subproblems can be tackled using neural CO methods like POMO to approximate the Pareto set. For a subproblem associated with a weight vector $\boldsymbol{\lambda}$, the solution construction process can be framed as a Markov decision process. Specifically, a solution is represented as a sequence $\boldsymbol{\pi} = \{\pi_1, \ldots, \pi_T\}$ of length $T$. For example, for the traveling salesman problem and capacitated vehicle routing problem, it denotes a tour of visited nodes. Similarly, for the knapsack problem, it signifies an order of selected items. Given an instance $s$, a stochastic policy $P(\boldsymbol{\pi}|\boldsymbol{\lambda}, s)$ sequentially generating solution $\boldsymbol{\pi}$ is defined as $P(\boldsymbol{\pi}|\boldsymbol{\lambda}, s) = \prod_{t=1}^{T} P_{\boldsymbol{\theta}}(\pi_t|\boldsymbol{\pi}_{1:t-1}, \boldsymbol{\lambda}, s)$, where the probability of node selection $P_{\boldsymbol{\theta}}(\pi_t|\boldsymbol{\pi}_{1:t-1}, \boldsymbol{\lambda}, s)$ is parameterized and governed by a deep model $\boldsymbol{\theta}$. The reward is quantified as the negative scalarized objective value, $-g(\boldsymbol{\pi}|\boldsymbol{\lambda}, s)$.

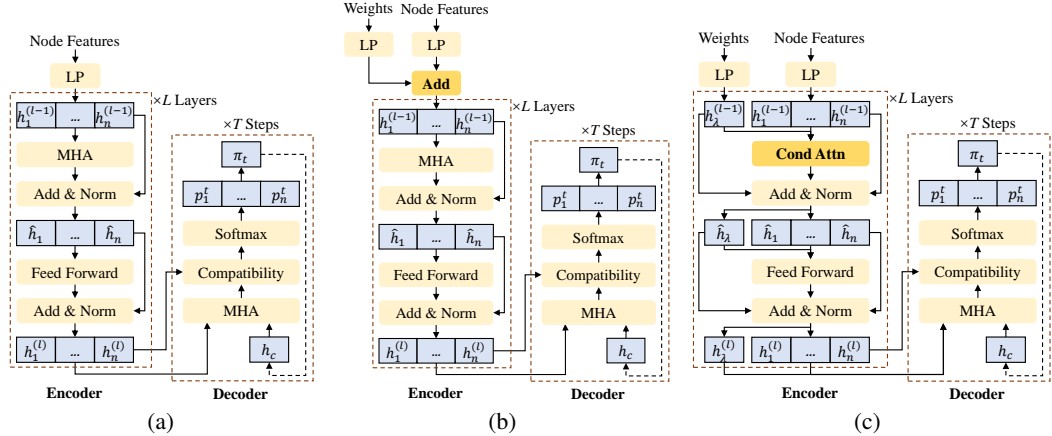

Figure 1: The architectures of the deep models. (a) Single-objective model, POMO. (b) Weight embedding with addition (Add). (c) Weight embedding with conditional attention (Cond Attn).

## 4 METHODOLOGY

To learn $P_{\boldsymbol{\theta}}(\pi_t|\boldsymbol{\pi}_{1:t-1}, \boldsymbol{\lambda}, s)$ in the stochastic policy for scalarized subproblems, which are associated with the weight, our weight embedding method neatly inputs $\boldsymbol{\lambda}$ to the single-objective model (as illustrated in Figure 1(a)). This method efficiently learns weight-specific representations, capturing and enhancing the weight-instance interaction. To demonstrate its potentials, we instantiate our method into two models. First, we introduce a succinct weight embedding model with addition (WE-Add), as depicted in Figure 1(b). Second, we design an enhanced weight embedding model with conditional attention (WE-CA), as depicted in Figure 1(c), which achieved new SOTA performance.

### 4.1 THE SUCCINCT MODEL: WEIGHT EMBEDDING WITH ADDITION

**Single-objective model.** The single-objective model follows the encoder-decoder structure, such as the prevailing POMO (Kwon et al., 2020). As presented in Figure 1(a), given a problem instance $s$ containing $n$ nodes with $Z$-dimensional features $\boldsymbol{v}_1, \dots, \boldsymbol{v}_n \in R^Z$ (see Appendix B), the initial node embeddings $\boldsymbol{h}_1^{(0)}, \dots, \boldsymbol{h}_n^{(0)} \in R^d$, in which $d$ is empirically set to 128 (Kool et al., 2019), are first derived via a linear projection (LP) with a trainable matrix $W^v \in R^{d \times Z}$ and bias $\boldsymbol{b}^v \in R^d$, as follows,

$$\boldsymbol{h}_i^{(0)} = W^v \boldsymbol{v}_i + \boldsymbol{b}^v, \forall i \in \{1, \dots, n\}. \tag{1}$$

In the encoder, the eventual node embeddings $\boldsymbol{h}_1^{(L)}, \dots, \boldsymbol{h}_n^{(L)}$ are then produced by going through $L = 6$ attention layers (Kwon et al., 2020). Each layer $l \in \{1, ..., L\}$ consists of a series of components in order: a multi-head attention (MHA) sublayer with 8 heads (Kool et al., 2019), a skip-connection (He et al., 2016) and instance normalization (IN) (Ulyanov et al., 2016) sublayer (Add & Norm), a fully connected feed-forward sublayer, and another Add & Norm sublayer, as follows,

$$\hat{\boldsymbol{h}}_i = \text{IN}(\boldsymbol{h}_i^{(l-1)} + \text{MHA}(\boldsymbol{h}_i^{(l-1)}, \{\boldsymbol{h}_1^{(l-1)}, \dots, \boldsymbol{h}_n^{(l-1)}\})), \forall i \in \{1, \dots, n\}, \tag{2}$$

$$\boldsymbol{h}_i^{(l)} = \text{IN}(\hat{\boldsymbol{h}}_i + \text{FF}(\hat{\boldsymbol{h}}_i)), \forall i \in \{1, \dots, n\}. \tag{3}$$

In the decoder, the node embeddings are used to autoregressively compute the probability of node selection with $T$ steps. At decoding step $t \in \{1, ..., T\}$, the *glimpse* $\boldsymbol{q}_c$ of *context* embedding $\boldsymbol{h}_c$ (see Appendix B) is produced by an MHA layer with 8 attention heads, and the *compatibility* $\boldsymbol{\alpha}$ is then computed, as follows,

$$\boldsymbol{q}_c = \text{MHA}(\boldsymbol{h}_c, \{\boldsymbol{h}_1^{(L)}, \dots, \boldsymbol{h}_n^{(L)}\}), \tag{4}$$

$$\alpha_i = \begin{cases} -\infty, & \text{node } i \text{ is masked} \\ C \cdot \tanh(\frac{\boldsymbol{q}_c^T(W^K \boldsymbol{h}_i^{(L)})}{\sqrt{d}}), & \text{otherwise} \end{cases} \tag{5}$$

where $C$ is set to 10 (Kool et al., 2019). Finally, the probability of node selection for a single-objective CO problem is calculated via softmax, i.e., $P_{\boldsymbol{\theta}}(\pi_t|\boldsymbol{\pi}_{1:t-1}, s) = \text{Softmax}(\boldsymbol{\alpha})$.

**Weight-specific node embeddings with addition.** To solve the scalarized subproblem related with the weight vector $\boldsymbol{\lambda}$, we input it to the model and directly add the produced initial weight embedding with the initial node embedding, as illustrated in Figure 1(b). Concretely, we only replace Equation (1) by Equation (6) below and keep the other parts of the model unchanged:

$$\boldsymbol{h}_i^{(0)} = (W^\lambda \boldsymbol{\lambda} + \boldsymbol{b}^\lambda) + (W^v \boldsymbol{v}_i + \boldsymbol{b}^v), \ \forall i \in \{1, \dots, n\}, \tag{6}$$

where the trainable parameters $W^\lambda \in R^{d \times M}$ and $\boldsymbol{b}^\lambda \in R^d$ are used for the initial weight embedding, while $W^v \in R^{d \times Z}$ and $\boldsymbol{b}^v \in R^d$ are employed for the initial node embeddings. In such a straightforward way, the weight-specific node embeddings are obtained. Subsequently, the weight and instance information interacts in the original encoder and decoder. Eventually, this process yields the probability of weight-specific node selection policy $P_{\boldsymbol{\theta}}(\pi_t|\boldsymbol{\pi}_{1:t-1}, \boldsymbol{\lambda}, s)$ for the scalarized subproblem. It is worth highlighting that this succinct weight embedding model with addition can already outperform most existing ones with complicated learning techniques, as shown in Table 1.

## 4.2 THE ENHANCED MODEL: WEIGHT EMBEDDING WITH CONDITIONAL ATTENTION

To more effectively capture the weight-instance interaction, we propose an enhanced weight embedding model with a conditional attention mechanism, as illustrated in Figure 1(c). In the conditional attention model, the weight embedding is incorporated into node embeddings in a feature-wise manner. Besides, the weight embedding and node embeddings are simultaneously updated to diminish disharmony of their interaction. As a result, it enables the model to more accurately discern their joint influence on the solution of the weight-specific subproblem.

**Conditional attention model.** The initial weight and node embeddings are first obtained by Equation (7) with the trainable parameters $W^\lambda$, $\boldsymbol{b}^\lambda$, $W^v$, and $\boldsymbol{b}^v$, as follows,

$$\boldsymbol{h}_\lambda^{(0)} = W^\lambda \boldsymbol{\lambda} + \boldsymbol{b}^\lambda, \ \boldsymbol{h}_i^{(0)} = W^v \boldsymbol{v}_i + \boldsymbol{b}^v, \ \forall i \in \{1, \dots, n\}. \tag{7}$$

In layer $l \in \{1, ..., L\}$ of the encoder, the weight and node embeddings are jointly updated via a conditional attention sublayer, which is composed of the conditional embedding mechanism and the MHA mechanism. First, the node embeddings conditioned on the weight embedding are produced by a feature-wise affine transformation (Perez et al., 2018), as follows,

$$\boldsymbol{\gamma} = W^\gamma \boldsymbol{h}_\lambda^{(l-1)}, \ \boldsymbol{\beta} = W^\beta \boldsymbol{h}_\lambda^{(l-1)}, \ \boldsymbol{h}_i' = \boldsymbol{\gamma} \circ \boldsymbol{h}_i^{(l-1)} + \boldsymbol{\beta}, \ \forall i \in \{1, \dots, n\}, \tag{8}$$

where $W^\gamma$ and $W^\beta$ are trainable matrices; $\circ$ is the element-wise multiplication, i.e., the $j$-th feature of the $i$-th node embedding $h_{i,j}' = \gamma_j \cdot h_{i,j}^{(l-1)} + \beta_j$. Then, the weight and node embeddings are updated via the MHA mechanism and an Add & Norm sublayer, as follows,

$$\hat{\boldsymbol{h}}_\lambda = \text{IN}(\boldsymbol{h}_\lambda^{(l-1)} + \text{MHA}(\boldsymbol{h}_\lambda^{(l-1)}, \{\boldsymbol{h}_\lambda^{(l-1)}, \boldsymbol{h}_1', \dots, \boldsymbol{h}_n'\})), \tag{9}$$

$$\hat{\boldsymbol{h}}_i = \text{IN}(\boldsymbol{h}_i^{(l-1)} + \text{MHA}(\boldsymbol{h}_i', \{\boldsymbol{h}_\lambda^{(l-1)}, \boldsymbol{h}_1', \dots, \boldsymbol{h}_n'\})), \ \forall i \in \{1, \dots, n\}. \tag{10}$$

Afterwards, a fully connected feed-forward sublayer and another Add & Norm sublayer, i.e., Equation (3), are employed to yield the weight embedding $\boldsymbol{h}_\lambda^{(l)}$ and node embeddings $\boldsymbol{h}_1^{(l)}, \dots, \boldsymbol{h}_n^{(l)}$. Finally, the decoder leverages the weight and node embeddings to produce the *glimpse* $\boldsymbol{q}_c$ by Equation (11), and the probability of weight-specific node selection $P_{\boldsymbol{\theta}}(\pi_t|\boldsymbol{\pi}_{1:t-1}, \boldsymbol{\lambda}, s)$ can be computed by Equation (5), which is used to construct the solution for the scalarized subproblem:

$$\boldsymbol{q}_c = \text{MHA}(\boldsymbol{h}_c, \{\boldsymbol{h}_\lambda^{(L)}, \boldsymbol{h}_1^{(L)}, \dots, \boldsymbol{h}_n^{(L)}\}). \tag{11}$$

## 4.3 TRAINING AND INFERENCE

To train the deep model, we adopt the REINFORCE algorithm with a shared baseline (Kwon et al., 2020) (see Appendix C for more details). We would like to note that most existing neural MOCO methods specifically train a model for each problem size and thus exhibit limited generalization across sizes. However, besides training size-specific models, our method, benefiting from the proposed weight embedding, can even train a single unified model achieving favorable cross-size generalization

capabilities by directly sampling instances of various sizes. During inference, when $N$ weight vectors are given, $N$ corresponding subproblems are solved to approximate the Pareto set. Furthermore, unlike existing neural MOCO methods that rely on complex learning techniques to handle weights, our neat weight embedding method facilitates the parallel processing of subproblems. This capability significantly enhances the speed of inference (see Appendix D for more details).

## 5 EXPERIMENTS

### 5.1 EXPERIMENTAL SETTINGS

**Problems.** We evaluate the proposed method on three classic MOCO problems that are studied in most neural MOCO literature, including the multi-objective traveling salesman problem (MOTSP) (Lust & Teghem, 2010), multi-objective capacitated vehicle routing problem (MOCVRP) (Zajac & Huber, 2021), and multi-objective knapsack problem (MOKP) (Ishibuchi et al., 2015) (see Appendix B for more details). Three common sizes are considered, i.e., $n = 20/50/100$ for MOTSP and MOCVRP, and $n = 50/100/200$ for MOKP.

**Hyperparameters.** We configure most hyperparameters based on prior works (Lin et al., 2022a; Kwon et al., 2020), and our method introduces no additional hyperparameters. The model is trained over 200 epochs, with each epoch containing 100,000 randomly sampled instances. The batch size $B$ is set to 64. The Adam (Kingma & Ba, 2015) optimizer with learning rate $10^{-4}$ and weight decay $10^{-6}$ is adopted. The $N$ weight vectors are generated using the Das & Dennis (1998) method, with $N$ set to 101 for $M = 2$ and 105 for $M = 3$.

**Baselines.** Our methods, including weight embedding with addition (**WE-Add**) and weight embedding with conditional attention (**WE-CA**), are compared with three classes of strong baseline methods. They all adopt the weighted sum (WS) scalarization for fair comparisons. (1) The state-of-the-art single-model neural MOCO methods, including **PMOCO** (Lin et al., 2022a), **MORAM** (Wang et al., 2024), and **CNH** (Fan et al., 2024), where CNH is specialized for size generalization using extra size embedding. (2) The multi-model neural MOCO methods, including **DRL-MOA** (Li et al., 2021), **MDRL** (Zhang et al., 2023c), and **EMNH** (Chen et al., 2023a). Particularly, DRL-MOA trains $N$ POMO models for $N$ subproblems, with 200 epochs for the first one and 5 epochs for each remaining one via parameter transfer. MDRL and EMNH both fine-tune $N$ POMO models from a pretrained meta-model that shares the same structure, with the training and fine-tuning settings following those in (Chen et al., 2023a). (3) The non-learnable methods, including widely used MOEAs and other strong heuristics. Specifically, **MOEA/D** (Zhang & Li, 2007) and **NSGA-II** (Deb et al., 2002), both implemented with 4,000 iterations, are representative decomposition-based and dominance-based MOEAs, respectively. **MOGLS** (Jaszkiewicz, 2002), with 4,000 iterations and 100 local search steps in each iteration, and **PPLS/D-C** (Shi et al., 2024), with 200 iterations, are both of MOEAs specialized for MOCO, using a 2-opt heuristic for MOTSP and MOCVRP, and a greedy transformation heuristic (Ishibuchi et al., 2015) for MOKP. **WS-LKH** and **WS-DP**, combining WS scalarization with the strong LKH (Helsgaun, 2000; Tinós et al., 2018) and dynamic programming (DP) solvers for decomposed subproblems, are used for MOTSP and MOKP, respectively. All methods are executed on a machine with an RTX 3090 GPU and an Intel Xeon 4216 CPU. Our code is publicly available[1].

**Metrics.** The widely used hypervolume (HV) indicator (Audet et al., 2021) is adopted here to evaluate the performance of MOCO methods (see Appendix E for the definition), where higher HV means better solution set. The average HV, gaps with respect to WE-CA-Aug, and total solving time for 200 instances are reported. The methods with "-Aug" represent results using instance augmentation (Lin et al., 2022a) (see Appendix F) to further improve performance. A Wilcoxon rank-sum test with 1% significance level is conducted. The best result and the ones without statistical significance are highlighted in **bold**, while the second-best result and the ones without statistical significance are highlighted in underline.

### 5.2 RESULTS AND ANALYSES

**Comparison of specialized training on each problem size.** The comparison results on bi-objective TSP (Bi-TSP), bi-objective CVRP (Bi-CVRP), bi-objective KP (Bi-KP), and tri-objective TSP

---

[1] https://github.com/bill-cjb/WE

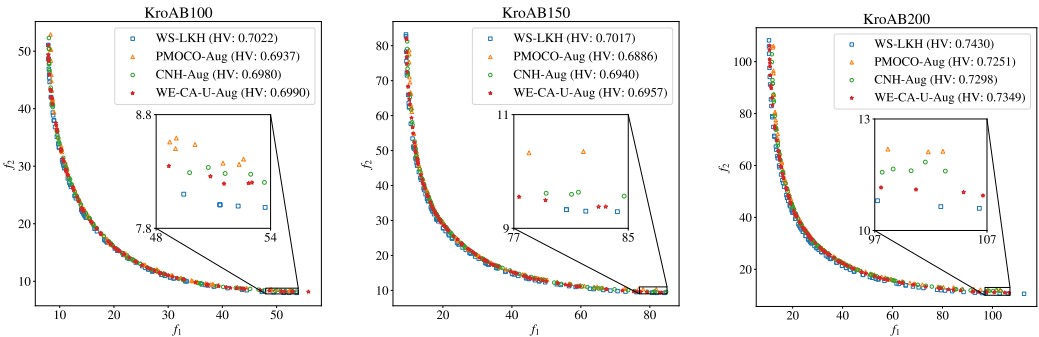

Figure 2: Pareto fronts on benchmark instances.

(Tri-TSP) are reported in Table 1. Our succinct WE-Add and enhanced WE-CA outperform most neural methods, where WE-CA is superior to WE-Add. Remarkably, compared with the state-of-the-art PMOCO, which also trains a single model, WE-CA acquires much smaller gaps in all cases, e.g., 0.24% vs 4.14% on Bi-CVRP100 and 1.19% vs 3.53% on Tri-TSP100. Compared with the multi-model methods, WE-CA gains much higher solution quality in most cases, except on Bi-KP100, Bi-KP200, Bi-CVRP20, and Bi-CVRP50, where it still presents competitive gaps (at most 0.06%). Especially, WE-CA, i.e., without instance augmentation, surpasses other neural baselines with instance augmentation on Bi-TSP100 and Tri-TSP100. Compared with the traditional non-learnable heuristics that consume much longer computational time due to the iterative search, such as WS-LKH taking 6.0 hours on Bi-TSP100, our WE-CA-Aug only takes 15 minutes while still achieving promising gaps.

**Generalization of unified training across problem sizes.** We train a unified model across problem sizes $n \in \{20, 21, \cdots, 100\}$ (except $n \in \{50, 51, \cdots, 200\}$ for Bi-KP) for WE-CA, CNH, and PMOCO. For MORAM, we use the provided pretrained model. The suffixes "-U" and "-$n$" are used to distinguish the models trained across various sizes and on a fixed size $n$, respectively. For WE-CA and PMOCO, only the results trained with "-50" are reported, as they are much better than that trained with "-20" and "-100". As shown in Table 2, among all the neural methods including CNH tailored for size generalization, WE-CA-U achieves the smallest average gap, which is even close to WE-CA specially trained on each size, i.e., the gap 0.00% on MOTSP, 0.02% on MOCVRP, and 0.03% on MOKP. On Tri-TSP and Bi-KP, CNH is even inferior to WE-CA-50 and WE-CA-100, respectively, which are both trained on a fixed size. Besides inferior solution quality, CNH also suffers from much longer solving and training time due to the extra size embedding. In addition, WE-CA-U manifests significant superiority to WE-CA-50, while PMOCO-U suffers from unstable training across sizes on some problems, like Bi-TSP and Bi-KP. We further assess the out-of-distribution generalization capability on the unseen larger sizes, i.e., Bi-TSP150 and Bi-TSP200. The results are reported in Table 3, where all the neural methods are trained on $n = 100$ except MORAM, CNH, and WE-CA-U. As shown, WE-CA-U significantly outperforms other neural methods and the classic MOEAs. Commendably, WE-CA-U is much more superior to CNH in the out-of-distribution case. Especially, WE-CA-U surpasses CNH-Aug equipped with instance augmentation on the larger size $n = 200$. We also test the methods on three commonly used instances adapted from TSPLIB (Reinelt, 1991), i.e., KroAB100, KroAB150, and KroAB200. The obtained Pareto fronts are visualized in Figure 2. Obviously, many solutions derived from WE-CA-U dominate those obtained by CNH and PMOCO, indicating that our method achieves smaller optimality gaps on decomposed subproblems. More generalization results are given in Appendix G.

**Pattern of weight-instance interaction.** On the one hand, we study the effect of weight-instance interaction via different parts of the model. Our WE-CA interacts weight information with instance information by the whole model, which is compared with WE-CA-Dec only inputting the weight information to the decoder for weight-instance interaction. Besides, we compare PMOCO with its two variants that employ the whole model for weight-instance interaction. They use a hypernetwork to learn the parameters of the encoder and whole model, named PMOCO-Enc and PMOCO-All, respectively. On the other hand, we study the effect of different weight embedding models. WE-CA

Table 1: Results of specialized training on each size with 200 random instances for MOCO problems.

| Method | | Bi-TSP20 | | | Bi-TSP50 | | | Bi-TSP100 | | |
| --- | --- | --- | --- | --- | --- | --- | --- | --- | --- | --- |
| | | HV↑ | Gap↓ | Time↓ | HV↑ | Gap↓ | Time↓ | HV↑ | Gap↓ | Time↓ |
| Non-learnable | WS-LKH | 0.6270 | 0.02% | 10m | **0.6415** | **-0.05%** | 1.8h | **0.7090** | **-0.33%** | 6.0h |
| | MOEA/D | 0.6241 | 0.48% | 1.7h | 0.6316 | 1.50% | 1.8h | 0.6899 | 2.38% | 2.2h |
| | NSGA-II | 0.6258 | 0.21% | 6.0h | 0.6120 | 4.55% | 6.1h | 0.6692 | 5.31% | 6.9h |
| | MOGLS | **0.6279** | **-0.13%** | 1.6h | 0.6330 | 1.28% | 3.7h | 0.6854 | 3.01% | 11h |
| | PPLS/D-C | 0.6256 | 0.24% | 26m | 0.6282 | 2.03% | 2.8h | 0.6844 | 3.16% | 11h |
| Multi-model | DRL-MOA | 0.6257 | 0.22% | 6s | 0.6360 | 0.81% | 9s | 0.6970 | 1.37% | 21s |
| | MDRL | 0.6271 | 0.00% | 5s | 0.6364 | 0.75% | 9s | 0.6969 | 1.39% | 17s |
| | EMNH | 0.6271 | 0.00% | 5s | 0.6364 | 0.75% | 9s | 0.6969 | 1.39% | 16s |
| Single-model | PMOCO | 0.6259 | 0.19% | 6s | 0.6351 | 0.95% | 10s | 0.6957 | 1.56% | 19s |
| | WE-Add | 0.6270 | 0.02% | 6s | 0.6386 | 0.41% | 10s | 0.7024 | 0.61% | 19s |
| | WE-CA | 0.6270 | 0.02% | 6s | 0.6391 | 0.33% | 10s | 0.7039 | 0.40% | 20s |
| Multi-model-Aug | MDRL-Aug | 0.6271 | 0.00% | 33s | 0.6408 | 0.06% | 1.7m | 0.7022 | 0.64% | 14m |
| | EMNH-Aug | 0.6271 | 0.00% | 33s | 0.6408 | 0.06% | 1.7m | 0.7023 | 0.62% | 14m |
| Single-model-Aug | PMOCO-Aug | 0.6270 | 0.02% | 1.1m | 0.6395 | 0.27% | 3.2m | 0.7016 | 0.72% | 15m |
| | WE-Add-Aug | 0.6272 | -0.02% | 1.1m | 0.6411 | 0.02% | 3.2m | 0.7058 | 0.13% | 15m |
| | WE-CA-Aug | 0.6271 | 0.00% | 1.1m | 0.6412 | 0.00% | 3.3m | 0.7067 | 0.00% | 15m |

| Method | | Bi-CVRP20 | | | Bi-CVRP50 | | | Bi-CVRP100 | | |
| --- | --- | --- | --- | --- | --- | --- | --- | --- | --- | --- |
| | | HV↑ | Gap↓ | Time↓ | HV↑ | Gap↓ | Time↓ | HV↑ | Gap↓ | Time↓ |
| Non-learnable | MOEA/D | 0.4255 | 1.07% | 2.3h | 0.4000 | 2.53% | 2.9h | 0.3953 | 3.16% | 5.0h |
| | NSGA-II | 0.4275 | 0.60% | 6.4h | 0.3896 | 5.07% | 8.8h | 0.3620 | 11.32% | 9.4h |
| | MOGLS | 0.4278 | 0.53% | 9.0h | 0.3984 | 2.92% | 20h | 0.3875 | 5.07% | 72h |
| | PPLS/D-C | 0.4287 | 0.33% | 1.6h | 0.4007 | 2.36% | 9.7h | 0.3946 | 3.33% | 38h |
| Multi-model | DRL-MOA | 0.4287 | 0.33% | 10s | 0.4076 | 0.68% | 12s | 0.4055 | 0.66% | 33s |
| | MDRL | 0.4291 | 0.23% | 8s | 0.4082 | 0.54% | 13s | 0.4056 | 0.64% | 32s |
| | EMNH | 0.4299 | 0.05% | 7s | 0.4098 | 0.15% | 13s | 0.4072 | 0.24% | 31s |
| Single-model | PMOCO | 0.4267 | 0.79% | 7s | 0.4036 | 1.66% | 12s | 0.3913 | 4.14% | 27s |
| | WE-Add | 0.4292 | 0.21% | 6s | 0.4089 | 0.37% | 12s | 0.4061 | 0.51% | 26s |
| | WE-CA | 0.4293 | 0.19% | 6s | 0.4090 | 0.34% | 12s | 0.4072 | 0.24% | 25s |
| Multi-model-Aug | MDRL-Aug | 0.4294 | 0.16% | 11s | 0.4092 | 0.29% | 36s | 0.4072 | 0.24% | 2.8m |
| | EMNH-Aug | **0.4302** | **-0.02%** | 11s | **0.4106** | **-0.05%** | 35s | 0.4079 | 0.07% | 2.8m |
| Single-model-Aug | PMOCO-Aug | 0.4294 | 0.16% | 14s | 0.4080 | 0.58% | 36s | 0.3969 | 2.77% | 2.7m |
| | WE-Add-Aug | 0.4300 | 0.02% | 14s | 0.4103 | 0.02% | 38s | 0.4079 | 0.07% | 2.5m |
| | WE-CA-Aug | 0.4301 | 0.00% | 14s | 0.4104 | 0.00% | 37s | **0.4082** | **0.00%** | 2.5m |

| Method | | Bi-KP50 | | | Bi-KP100 | | | Bi-KP200 | | |
| --- | --- | --- | --- | --- | --- | --- | --- | --- | --- | --- |
| | | HV↑ | Gap↓ | Time↓ | HV↑ | Gap↓ | Time↓ | HV↑ | Gap↓ | Time↓ |
| Non-learnable | WS-DP | **0.3561** | **0.00%** | 22m | 0.4532 | 0.02% | 2.0h | 0.3601 | 0.00% | 5.8h |
| | MOEA/D | 0.3540 | 0.59% | 1.6h | 0.4508 | 0.55% | 1.7h | 0.3581 | 0.56% | 1.8h |
| | NSGA-II | 0.3547 | 0.39% | 7.8h | 0.4520 | 0.29% | 8.0h | 0.3590 | 0.31% | 8.4h |
| | MOGLS | 0.3540 | 0.59% | 5.8h | 0.4510 | 0.51% | 10h | 0.3582 | 0.53% | 18h |
| | PPLS/D-C | 0.3528 | 0.93% | 18m | 0.4480 | 1.17% | 47m | 0.3541 | 1.67% | 1.5h |
| Multi-model | DRL-MOA | 0.3559 | 0.06% | 9s | 0.4531 | 0.04% | 18s | 0.3601 | 0.00% | 1.0m |
| | MDRL | 0.3530 | 0.87% | 6s | 0.4532 | 0.02% | 21s | 0.3601 | 0.00% | 1.2m |
| | EMNH | **0.3561** | **0.00%** | 6s | **0.4535** | **-0.04%** | 21s | **0.3603** | **-0.06%** | 1.2m |
| Single-model | PMOCO | 0.3552 | 0.25% | 9s | 0.4523 | 0.22% | 19s | 0.3595 | 0.17% | 1.3m |
| | WE-Add | 0.3558 | 0.08% | 8s | 0.4530 | 0.07% | 20s | 0.3600 | 0.03% | 1.1m |
| | WE-CA | **0.3561** | **0.00%** | 9s | 0.4533 | 0.00% | 19s | 0.3601 | 0.00% | 1.1m |

| Method | | Tri-TSP20 | | | Tri-TSP50 | | | Tri-TSP100 | | |
| --- | --- | --- | --- | --- | --- | --- | --- | --- | --- | --- |
| | | HV↑ | Gap↓ | Time↓ | HV↑ | Gap↓ | Time↓ | HV↑ | Gap↓ | Time↓ |
| Non-learnable | WS-LKH | **0.4712** | **0.00%** | 12m | **0.4440** | **-0.20%** | 1.9h | **0.5076** | **-0.79%** | 6.6h |
| | MOEA/D | 0.4702 | 0.21% | 1.9h | 0.4314 | 2.64% | 2.2h | 0.4511 | 10.42% | 2.4h |
| | NSGA-II | 0.4238 | 10.06% | 7.1h | 0.2858 | 35.50% | 7.5h | 0.2824 | 43.92% | 9.0h |
| | MOGLS | 0.4701 | 0.23% | 1.5h | 0.4211 | 4.97% | 4.1h | 0.4254 | 15.53% | 13h |
| | PPLS/D-C | 0.4698 | 0.30% | 1.4h | 0.4174 | 5.80% | 3.9h | 0.4376 | 13.11% | 14h |
| Multi-model | DRL-MOA | 0.4699 | 0.28% | 6s | 0.4303 | 2.89% | 9s | 0.4806 | 4.57% | 19s |
| | MDRL | 0.4699 | 0.28% | 5s | 0.4317 | 2.57% | 9s | 0.4852 | 3.65% | 16s |
| | EMNH | 0.4699 | 0.28% | 5s | 0.4324 | 2.41% | 9s | 0.4866 | 3.38% | 16s |
| Single-model | PMOCO | 0.4693 | 0.40% | 5s | 0.4315 | 2.62% | 8s | 0.4858 | 3.53% | 18s |
| | WE-Add | 0.4705 | 0.15% | 6s | 0.4379 | 1.17% | 10s | 0.4943 | 1.85% | 20s |
| | WE-CA | 0.4707 | 0.11% | 5s | 0.4389 | 0.95% | 9s | 0.4976 | 1.19% | 19s |
| Multi-model-Aug | MDRL-Aug | **0.4712** | **0.00%** | 2.6m | 0.4408 | 0.52% | 25m | 0.4958 | 1.55% | 1.7h |
| | EMNH-Aug | **0.4712** | **0.00%** | 2.6m | 0.4418 | 0.29% | 25m | 0.4973 | 1.25% | 1.7h |
| Single-model-Aug | PMOCO-Aug | **0.4712** | **0.00%** | 5.1m | 0.4409 | 0.50% | 28m | 0.4956 | 1.59% | 1.7h |
| | WE-Add-Aug | **0.4712** | **0.00%** | 5.1m | 0.4429 | 0.05% | 29m | 0.5016 | 0.40% | 1.8h |
| | WE-CA-Aug | **0.4712** | **0.00%** | 5.2m | 0.4431 | 0.00% | 29m | 0.5036 | 0.00% | 1.7h |

Table 2: Results of unified training across sizes with 200 random instances for MOCO problems.

| Method | Bi-TSP20 HV↑ | Gap↓ | Time↓ | Bi-TSP50 HV↑ | Gap↓ | Time↓ | Bi-TSP100 HV↑ | Gap↓ | Time↓ | Bi-TSP Avg. Gap↓ |
|---|---|---|---|---|---|---|---|---|---|---|
| MORAM | 0.6216 | 0.88% | 1s | 0.6255 | 2.45% | 2s | 0.6821 | 3.48% | 3s | 2.27% |
| CNH | 0.6270 | 0.02% | 14s | 0.6387 | 0.39% | 17s | 0.7019 | 0.68% | 29s | 0.36% |
| PMOCO-50 | 0.6262 | 0.14% | 6s | 0.6351 | 0.95% | 10s | 0.6915 | 2.15% | 19s | 1.08% |
| PMOCO-U | 0.6111 | 2.55% | 7s | 0.5939 | 7.38% | 11s | 0.6417 | 9.20% | 21s | 6.38% |
| WE-CA-50 | 0.6267 | 0.06% | 6s | 0.6391 | 0.33% | 10s | 0.6988 | 1.12% | 19s | 0.50% |
| WE-CA-U | 0.6270 | 0.02% | 7s | 0.6392 | 0.31% | 10s | 0.7034 | 0.47% | 21s | 0.26% |
| CNH-Aug | 0.6271 | 0.00% | 1.5m | 0.6410 | 0.03% | 4.1m | 0.7054 | 0.18% | 16m | 0.07% |
| PMOCO-50-Aug | 0.6270 | 0.02% | 1.0m | 0.6395 | 0.27% | 3.2m | 0.6977 | 1.27% | 15m | 0.52% |
| PMOCO-U-Aug | 0.6253 | 0.29% | 1.0m | 0.6126 | 4.46% | 3.3m | 0.6558 | 7.20% | 15m | 3.98% |
| WE-CA-50-Aug | **0.6272** | **-0.02%** | 1.0m | 0.6412 | 0.00% | 3.3m | 0.7034 | 0.47% | 15m | 0.15% |
| WE-CA-U-Aug | 0.6271 | 0.00% | 1.0m | **0.6413** | **-0.02%** | 3.3m | **0.7066** | **0.01%** | 16m | **0.00%** |

| Method | Bi-CVRP20 HV↑ | Gap↓ | Time↓ | Bi-CVRP50 HV↑ | Gap↓ | Time↓ | Bi-CVRP100 HV↑ | Gap↓ | Time↓ | Bi-CVRP Avg. Gap↓ |
|---|---|---|---|---|---|---|---|---|---|---|
| CNH | 0.4287 | 0.33% | 15s | 0.4087 | 0.41% | 17s | 0.4065 | 0.42% | 31s | 0.39% |
| PMOCO-50 | 0.4191 | 2.56% | 9s | 0.4036 | 1.66% | 12s | 0.4014 | 1.67% | 26s | 1.96% |
| PMOCO-U | 0.4275 | 0.60% | 8s | 0.4068 | 0.88% | 13s | 0.4044 | 0.93% | 25s | 0.80% |
| WE-CA-50 | 0.4238 | 1.46% | 7s | 0.4090 | 0.34% | 12s | 0.4025 | 1.40% | 25s | 1.07% |
| WE-CA-U | 0.4290 | 0.26% | 7s | 0.4089 | 0.37% | 12s | 0.4068 | 0.34% | 26s | 0.32% |
| CNH-Aug | 0.4299 | 0.05% | 22s | 0.4101 | 0.07% | 45s | 0.4077 | 0.12% | 2.5m | 0.08% |
| PMOCO-50-Aug | 0.4270 | 0.72% | 15s | 0.4080 | 0.58% | 36s | 0.4051 | 0.76% | 2.4m | 0.69% |
| PMOCO-U-Aug | 0.4296 | 0.12% | 15s | 0.4095 | 0.22% | 40s | 0.4071 | 0.27% | 2.4m | 0.20% |
| WE-CA-50-Aug | 0.4277 | 0.56% | 15s | **0.4104** | **0.00%** | 37s | 0.4056 | 0.64% | 2.5m | 0.40% |
| WE-CA-U-Aug | **0.4300** | **0.02%** | 14s | 0.4103 | 0.02% | 40s | **0.4081** | **0.02%** | 2.5m | **0.02%** |

| Method | Bi-KP50 HV↑ | Gap↓ | Time↓ | Bi-KP100 HV↑ | Gap↓ | Time↓ | Bi-KP200 HV↑ | Gap↓ | Time↓ | Bi-KP Avg. Gap↓ |
|---|---|---|---|---|---|---|---|---|---|---|
| CNH | 0.3556 | 0.14% | 18s | 0.4527 | 0.13% | 27s | 0.3598 | 0.08% | 1.2m | 0.12% |
| PMOCO-100 | 0.3548 | 0.37% | 11s | 0.4523 | 0.22% | 19s | 0.3527 | 2.05% | 1.0m | 0.88% |
| PMOCO-U | 0.3503 | 1.63% | 10s | 0.4484 | 1.08% | 19s | 0.3536 | 1.81% | 1.0m | 1.50% |
| WE-CA-100 | **0.3559** | **0.06%** | 10s | **0.4533** | **0.00%** | 19s | 0.3591 | 0.28% | 1.0m | 0.11% |
| WE-CA-U | 0.3558 | 0.08% | 9s | 0.4531 | 0.04% | 21s | **0.3602** | **-0.03%** | 1.1m | **0.03%** |

| Method | Tri-TSP20 HV↑ | Gap↓ | Time↓ | Tri-TSP50 HV↑ | Gap↓ | Time↓ | Tri-TSP100 HV↑ | Gap↓ | Time↓ | Tri-TSP Avg. Gap↓ |
|---|---|---|---|---|---|---|---|---|---|---|
| MORAM | 0.4573 | 2.95% | 1s | 0.4101 | 7.45% | 2s | 0.4588 | 8.90% | 3s | 6.43% |
| CNH | 0.4698 | 0.30% | 10s | 0.4358 | 1.65% | 14s | 0.4931 | 2.08% | 26s | 1.34% |
| PMOCO-50 | 0.4682 | 0.64% | 6s | 0.4315 | 2.62% | 8s | 0.4824 | 4.21% | 21s | 2.49% |
| PMOCO-U | 0.4691 | 0.45% | 7s | 0.4318 | 2.55% | 10s | 0.4873 | 3.24% | 21s | 2.08% |
| WE-CA-50 | 0.4691 | 0.45% | 5s | 0.4389 | 0.95% | 9s | 0.4877 | 3.16% | 20s | 1.52% |
| WE-CA-U | 0.4707 | 0.11% | 5s | 0.4389 | 0.95% | 9s | 0.4975 | 1.21% | 20s | 0.76% |
| CNH-Aug | 0.4704 | 0.17% | 8.0m | 0.4409 | 0.50% | 33m | 0.4996 | 0.79% | 2.1h | 0.49% |
| PMOCO-50-Aug | **0.4713** | **-0.02%** | 5.2m | 0.4409 | 0.50% | 28m | 0.4933 | 2.05% | 1.7h | 0.84% |
| PMOCO-U-Aug | 0.4712 | 0.00% | 5.2m | 0.4406 | 0.56% | 31m | 0.4968 | 1.35% | 1.7h | 0.64% |
| WE-CA-50-Aug | 0.4711 | 0.02% | 5.3m | 0.4431 | 0.00% | 29m | 0.4991 | 0.89% | 1.9h | 0.30% |
| WE-CA-U-Aug | 0.4712 | 0.00% | 5.2m | **0.4432** | **-0.02%** | 31m | **0.5035** | **0.02%** | 1.8h | **0.00%** |

is compared with the addition model (WE-Add), conditional embedding model (WE-CA w/o A), and attention model (WE-CA w/o C). All results are presented in Appendix H, and we observe that WE-CA achieves the best performance, which verified our design for weight-instance interaction.

**Optimality on scalarized subproblems.** To study the optimality on decomposed subproblems, the scalarized objective values of three representative subproblems associated with $\lambda = (1, 0)$, $(0.5, 0.5)$, and $(0, 1)$ on Bi-TSP100 are recorded in Table 4. The single-objective methods, LKH and POMO, specialized on the three subproblems are also included. Among the neural MOCO methods, WE-CA achieves the smallest gaps on all three subproblems. Compared with four weight embedding methods, PMOCO, EMNH, and MDRL all yield significant gaps, especially on the two extreme subproblems. WE-CA w/o A displays the superiority on the two extreme subproblems, while WE-CA w/o C exhibits the close performance on all subproblems. Beyond them, our WE-CA effectively combines their advantages on all subproblems, resulting in the best overall performance on the MOCO problems.

**Unnecessariness of problem size embedding for size generalization.** CNH uses an extra size-aware decoder to learn the problem size embedding (PSE) by the sinusoidal positional encoding. However, PSE is unscalable when the problem size is larger than the predefined maximum of the sinusoidal positional encoding (we set 300, 200, 300, and 200 for Bi-TSP, Bi-CVRP, Bi-KP, and Tri-TSP,

Table 3: Out-of-distribution generalization results on 200 random instances for larger-size problems.

| Method | Bi-TSP150 | | | Bi-TSP200 | | |
|---|---|---|---|---|---|---|
| | HV↑ | Gap↓ | Time↓ | HV↑ | Gap↓ | Time↓ |
| WS-LKH | **0.7149** | **-1.71%** | 13h | **0.7490** | **-2.50%** | 22h |
| MOEA/D | 0.6809 | 3.13% | 2.4h | 0.7139 | 2.30% | 2.7h |
| NSGA-II | 0.6659 | 5.26% | 6.8h | 0.7045 | 3.59% | 6.9h |
| MOGLS | 0.6768 | 3.71% | 22h | 0.7114 | 2.64% | 38h |
| PPLS/D-C | 0.6784 | 3.49% | 21h | 0.7106 | 2.75% | 32h |
| MORAM | 0.6711 | 4.52% | 4s | 0.7024 | 3.87% | 7s |
| DRL-MOA | 0.6901 | 1.82% | 45s | 0.7219 | 1.20% | 1.5m |
| MDRL | 0.6922 | 1.52% | 40s | 0.7251 | 0.77% | 1.4m |
| EMNH | 0.6930 | 1.41% | 40s | 0.7260 | 0.64% | 1.4m |
| PMOCO | 0.6910 | 1.69% | 50s | 0.7231 | 1.04% | 1.5m |
| WE-CA | 0.6991 | 0.54% | 56s | 0.7264 | 0.59% | 1.6m |
| CNH | 0.6985 | 0.63% | 1.1m | 0.7292 | 0.21% | 1.9m |
| WE-CA-U | 0.7008 | 0.30% | 57s | 0.7346 | -0.53% | 1.6m |
| MDRL-Aug | 0.6976 | 0.75% | 47m | 0.7299 | 0.11% | 1.6h |
| EMNH-Aug | 0.6983 | 0.65% | 47m | 0.7307 | 0.00% | 1.6h |
| PMOCO-Aug | 0.6967 | 0.88% | 47m | 0.7283 | 0.33% | 1.6h |
| WE-CA-Aug | 0.7029 | 0.00% | 51m | 0.7307 | 0.00% | 1.7h |
| CNH-Aug | 0.7025 | 0.06% | 52m | 0.7343 | -0.49% | 1.7h |
| WE-CA-U-Aug | 0.7044 | -0.21% | 50m | 0.7381 | -1.01% | 1.7h |

Table 4: The results of scalarized objectives on 200 random instances for various subproblems.

| Method | Bi-TSP100 | | | | | |
|---|---|---|---|---|---|---|
| | $g(\boldsymbol{\pi}|(1,0))$↓ | Gap↓ | $g(\boldsymbol{\pi}|(0.5,0.5))$↓ | Gap↓ | $g(\boldsymbol{\pi}|(0,1))$↓ | Gap↓ |
| LKH | **7.7632** | **0.00%** | **17.3094** | **0.00%** | **7.7413** | **0.00%** |
| POMO-Aug | 7.7659 | 0.03% | 17.4421 | 0.77% | 7.7716 | 0.39% |
| MDRL-Aug | 8.0316 | 3.46% | 17.6209 | 1.80% | 8.0290 | 3.72% |
| EMNH-Aug | 8.0620 | 3.85% | 17.5979 | 1.67% | 8.0439 | 3.91% |
| PMOCO-Aug | 8.1401 | 4.85% | 17.5723 | 1.52% | 8.1202 | 4.89% |
| WE-Add-Aug | 7.8389 | 0.98% | 17.5228 | 1.23% | 7.8130 | 0.93% |
| WE-CA w/o A-Aug | 7.8202 | 0.73% | 17.5271 | 1.26% | 7.7939 | 0.68% |
| WE-CA w/o C-Aug | 7.8417 | 1.01% | 17.4847 | 1.01% | 7.8123 | 0.92% |
| WE-CA-Aug | 7.8198 | 0.73% | 17.4633 | 0.89% | 7.7937 | 0.68% |

Table 5: Effect of problem size embedding on cross-size generalization.

| Method | Bi-TSP20 | | | Bi-TSP50 | | | Bi-TSP100 | | | Bi-TSP | |
|---|---|---|---|---|---|---|---|---|---|---|---|
| | HV↑ | Gap↓ | Time↓ | HV↑ | Gap↓ | Time↓ | HV↑ | Gap↓ | Time↓ | Avg. Gap↓ | Training time↓ |
| CNH | 0.6270 | 0.02% | 14s | 0.6387 | 0.39% | 17s | 0.7019 | 0.68% | 29s | 0.36% | 30h |
| WE-CA-PSE-Enc | 0.6270 | 0.02% | 13s | 0.6392 | 0.31% | 16s | 0.7033 | 0.48% | 29s | 0.27% | 30h |
| WE-CA-PSE-Dec | 0.6270 | 0.02% | 14s | 0.6390 | 0.34% | 18s | 0.7030 | 0.52% | 30s | 0.29% | 30h |
| WE-CA-U | 0.6270 | 0.02% | 7s | 0.6392 | 0.31% | 10s | 0.7034 | 0.47% | 21s | 0.26% | 19h |

respectively). Moreover, PSE results in much more solving and training time, as reported in Table 5. We also apply PSE to our WE-CA in two ways, i.e., injecting PSE into the decoder as the same with CNH and into the encoder as a variant, denoted as WE-CA-PSE-Dec and WE-CA-PSE-Enc, respectively. However, PSE slightly damages the performance of WE-CA, and also considerably increases solving and training time. This is because our method inherently possesses cross-size generalization capability learned directly from data. These results indicate that PSE is unnecessary for our weight embedding method, again underscoring the elegance and superiority of our WE.

## 6 CONCLUSION

This paper proposes a neat weight embedding method to effectively solve MOCO problems. By directly learning weight-specific representations, our method captures crucial weight-instance interactions, thereby reducing the optimality gaps. Extensive results on MOTSP, MOCVRP, and MOKP showed that our method surpasses the state-of-the-art neural methods and even manifests favored cross-size generalization capability. We acknowledge certain limitations: To effectively address real-world MOCO problems with complex constraints, this method may require further integration with appropriate constraint-handling techniques. Besides, the application of more sophisticated sampling and learning techniques across different problem sizes could further improve generalization.

ACKNOWLEDGMENTS AND DISCLOSURE OF FUNDING

This work is supported by the National Natural Science Foundation of China (62472461), and the Guangdong Basic and Applied Basic Research Foundation (2025A1515010129).

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

## A  SUMMARY OF THE DECOMPOSITION-BASED NEURAL MOCO METHODS

The characteristics of the decomposition-based neural MOCO methods are summarized in Table 6. Our weight embedding methods, which yield new state-of-the-art performance, adopt neat extra techniques beyond the single-objective model to handle all subproblems. Specifically, our succinct WE-Add only employs the addition with negligible complexity, which brings the smallest number of model parameters yet outperforms most neural methods. Our enhanced WE-CA couples the conditional embedding mechanism with the attention mechanism, achieving the superior performance with the smaller number of model parameters compared with the state-of-the-art neural methods.

## B  DETAILS OF MOCO PROBLEMS

### B.1  MULTI-OBJECTIVE TRAVELING SALESMAN PROBLEM

**Problem statement**  For the multi-objective traveling salesman problem (MOTSP) with $n$ nodes, node $i \in \{1, ..., n\}$ has $M$ groups of 2-dimensional coordinates, where $M$ is the number of objectives. The $m$-th Euclidean distance $c_{ij}^m$ between node $i$ and $j$ is calculated by their $m$-th coordinate. The goal is to find a solution $\boldsymbol{\pi}$ visiting all nodes to minimize all the $M$ total distances simultaneously, i.e., $\min \boldsymbol{f}(\boldsymbol{\pi}) = (f_1(\boldsymbol{\pi}), f_2(\boldsymbol{\pi}), \ldots, f_M(\boldsymbol{\pi}))$, where $f_m(\boldsymbol{\pi}) = c_{\pi_n,\pi_1}^m + \sum_{i=1}^{n-1} c_{\pi_i,\pi_{i+1}}^m, \forall m \in \{1, \ldots, M\}$.

**Instance generation**  For the $M$-objective TSP, the coordinates of each instance are sampled from uniform distribution on $[0, 1]^{2M}$.

**Node features and context embedding**  The inputs of the $M$-objective TSP are $n$ nodes with $2M$-dimensional features. At each decoding step, the context embedding $\boldsymbol{h}_c$ is defined as the concatenation of the embedding of the first visited node and the embedding of the last visited node. All the visited nodes are masked.

### B.2  MULTI-OBJECTIVE CAPACITATED VEHICLE ROUTING PROBLEM

**Problem statement**  We consider bi-objective capacitated vehicle routing problem (Bi-CVRP) with $n$ customer nodes and a depot node. Each node is featured by a 2-dimensional coordinate, and each customer node involves a demand. A fleet of homogeneous vehicles with identical capacity initialized at the depot must serve all the customers and finally return to the depot. The remaining capacity of vehicles must be no less than the demand of the customer when serving each customer. The two conflicting objectives are the total tour length and the makespan, i.e., the length of the longest route.

**Instance generation**  For Bi-CVRP, the coordinates of the depot and customers are sampled from uniform distribution on $[0, 1]^2$. The demand is uniformly sampled from $\{1, \ldots, 9\}$. The vehicle capacity is set as 30, 40, and 50 for $20 \leq n < 40$, $40 \leq n < 70$, and $70 \leq n \leq 100$, respectively. Without loss of generality, the demands are normalized by the capacity.

**Node features and context embedding**  The inputs of Bi-CVRP are $n$ customer nodes with 3-dimensional features and a depot node with 2-dimensional features. Their initial embeddings are

Table 6: Summary of the decomposition-based neural MOCO methods.

| Method | Extra techniques for subproblems | Flexibility | #(Parameters) | Performance |
|--------|----------------------------------|-------------|---------------|-------------|
| DRL-MOA | Transfer learning | Multi-model | 133.37M | large gap |
| MDRL | Meta learning | Multi-model | 133.37M | medium gap |
| EMNH | Meta learning | Multi-model | 133.37M | medium gap |
| MORAM | Weight router | Single-model | 1.30M | large gap |
| PMOCO | Hypernetwork | Single-model | 1.50M | large gap |
| CNH | Size-aware decoder | Single-model | 1.63M | small gap, good generalization |
| WE-Add | Addition | Single-model | 1.27M | small gap |
| WE-CA | Feature-wise linear projection | Single-model | 1.47M | smallest gap, best generalization |

---

**Algorithm 1** Training algorithm

---

1: **Input:** weight distribution $\Lambda$, instance distribution $\mathcal{S}_n$ on problem size $n$, number of training steps $E$, batch size $B$
2: Initialize the model parameters $\boldsymbol{\theta}$
3: **for** $e = 1$ to $E$ **do**
4:     **if** training a specialized model on a fixed size $n$ **then**
5:        $s_i \sim$ **SampleInstance**$(\mathcal{S}_n)$    $\forall i \in \{1, \cdots, B\}$
6:     **else if** training a unified model across various sizes **then**
7:        $n \sim$ **SampleSize**$(\mathcal{N}), \; s_i \sim$ **SampleInstance**$(\mathcal{S}_n)$    $\forall i \in \{1, \cdots, B\}$
8:     **end if**
9:     $\boldsymbol{\lambda} \sim$ **SampleWeight**$(\Lambda)$
10:     $\boldsymbol{\pi}_i^j \sim$ **SampleSolution**$(P(\boldsymbol{\pi}|\boldsymbol{\lambda}, s_i))$    $\forall i \in \{1, \cdots, B\}$    $\forall j \in \{1, \cdots, n\}$
11:     $b_i \leftarrow \frac{1}{n} \sum_{j=1}^{n} g(\boldsymbol{\pi}_i^j | \boldsymbol{\lambda}, s_i)$    $\forall i \in \{1, \cdots, B\}$
12:     $\nabla \mathcal{J}(\boldsymbol{\theta}) \leftarrow \frac{1}{Bn} \sum_{i=1}^{B} \sum_{j=1}^{n} [(g(\boldsymbol{\pi}_i^j | \boldsymbol{\lambda}, s_i) - b_i) \nabla_{\boldsymbol{\theta}} \log P(\boldsymbol{\pi}_i^j | \boldsymbol{\lambda}, s_i)]$
13:     $\boldsymbol{\theta} \leftarrow$ **Adam**$(\boldsymbol{\theta}, \nabla \mathcal{J}(\boldsymbol{\theta}))$
14: **end for**
15: **Output:** The model parameter $\boldsymbol{\theta}$

---

derived by two separate linear projections. At each decoding step, the context embedding $\boldsymbol{h}_c$ is defined as the concatenation of the embedding of the last visited node and the remaining vehicle capacity. All the visited nodes and those with a larger demand than the remaining vehicle capacity are masked.

### B.3 MULTI-OBJECTIVE KNAPSACK PROBLEM

**Problem statement** For the multi-objective knapsack problem (MOKP) with $M$ objectives and $n$ items, each item involves a weight and $M$ separate values. The goal is to find a solution to maximize all the $M$ objectives simultaneously without violating the knapsack capacity.

**Instance generation** The values and weight of each item are all sampled from uniform distribution on $[0, 1]$. The knapsack capacity is set as 12.5 and 25 for $50 \leq n < 100$ and $100 \leq n \leq 200$, respectively.

**Node features and context embedding** The inputs of Bi-KP are $n$ nodes (i.e., items) with 3-dimensional features. At each decoding step, the context embedding $\boldsymbol{h}_c$ is defined as the concatenation of the graph embedding $\bar{\boldsymbol{h}} = \sum_{i=1}^{n} \boldsymbol{h}_i / n$ and the remaining knapsack capacity. All the selected items and those with a larger weight than the remaining knapsack capacity are masked.

## C TRAINING ALGORITHM

The training algorithm is provided in Algorithm 1. To train a unified model with cross-size generalization capability, we sample instances from various problem sizes (as in Line 7).

## D PARALLEL INFERENCE FOR SUBPROBLEMS

Most of the existing neural MOCO methods learn weight-specific model parameters for the corresponding subproblems. It is difficult for these methods to implement parallel solving of subproblems, since they need to handle a batch of $N$ deep neural networks in parallel. By contrast, our method directly inputs the weight vector into the deep model and learns weight-specific representations, which can easily tackle a batch of $N$ weight vectors in mainstream deep learning frameworks such as Pytorch. The total runtime for solving all subproblems in parallel is reported in Table 7. As shown, parallel inference for subproblems can curtail the solving time. Notably, the solving speed is not accelerated to $N$ times actually, because it also depends on the memory capacity and parallel computing capability.

Table 7: Running time of solving subproblems in parallel.

| Method | Bi-TSP20 | Bi-TSP50 | Bi-TSP100 | Bi-CVRP20 | Bi-CVRP50 | Bi-CVRP100 | KroAB100 | KroAB150 | KroAB200 |
|---|---|---|---|---|---|---|---|---|---|
| WE-CA | 6s | 10s | 20s | 6s | 12s | 25s | 9s | 19s | 25s |
| WE-CA (parallel) | 2s | 4s | 15s | 3s | 6s | 19s | 1s | 1s | 2s |

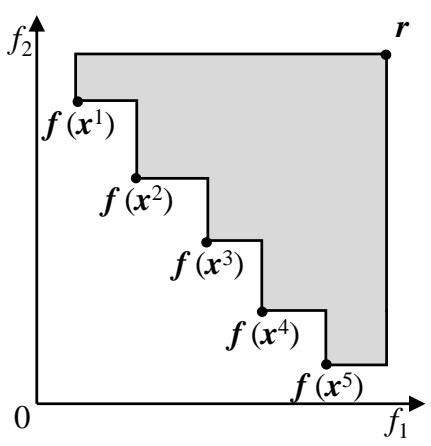

Figure 3: Hypervolume illustration.

Table 8: Reference points and ideal points.

| Problem | Size | $r$ | $z$ |
|---|---|---|---|
| Bi-TSP | 20 | (20, 20) | (0, 0) |
| | 50 | (35, 35) | (0, 0) |
| | 100 | (65, 65) | (0, 0) |
| | 150 | (85, 85) | (0, 0) |
| | 200 | (115, 115) | (0, 0) |
| Bi-CVRP | 20 | (30, 4) | (0, 0) |
| | 50 | (45, 4) | (0, 0) |
| | 100 | (80, 4) | (0, 0) |
| Bi-KP | 50 | (5, 5) | (30, 30) |
| | 100 | (20, 20) | (50, 50) |
| | 200 | (30, 30) | (75, 75) |
| Tri-TSP | 20 | (20, 20, 20) | (0, 0, 0) |
| | 50 | (35, 35, 35) | (0, 0, 0) |
| | 100 | (65, 65, 65) | (0, 0, 0) |

## E  HYPERVOLUME INDICATOR

Hypervolume (HV) is a mainstream indicator to evaluate the performance of MOCO methods, as it can comprehensively measure the optimality and diversity of the obtained Pareto front without the ground truth. $\text{HV}_{\boldsymbol{r}}(\mathcal{F})$ of a Pareto front $\mathcal{F}$ with respect to a reference point $\boldsymbol{r} \in \mathcal{R}^M$ is defined as follows,

$$\text{HV}_{\boldsymbol{r}}(\mathcal{F}) = \mu \left( \bigcup_{\boldsymbol{f}(\boldsymbol{x}) \in \mathcal{F}} [\boldsymbol{f}(\boldsymbol{x}), \boldsymbol{r}] \right), \tag{12}$$

where $\mu$ is the Lebesgue measure, and $[\boldsymbol{f}(\boldsymbol{x}), \boldsymbol{r}]$ is an $M$-dimensional cube, i.e., $[\boldsymbol{f}(\boldsymbol{x}), \boldsymbol{r}] = [f_1(\boldsymbol{x}), r_1] \times \cdots \times [f_M(\boldsymbol{x}), r_M]$. A 2-dimensional example of a Pareto front with five solutions is demonstrated in Figure 3, where $\mathcal{F} = \{\boldsymbol{f}(\boldsymbol{x}^1), \boldsymbol{f}(\boldsymbol{x}^2), \boldsymbol{f}(\boldsymbol{x}^3), \boldsymbol{f}(\boldsymbol{x}^4), \boldsymbol{f}(\boldsymbol{x}^5)\}$. $\text{HV}_{\boldsymbol{r}}(\mathcal{F})$ is equal to the size of the gray area.

HV is finally normalized as $\text{HV}'_{\boldsymbol{r}}(\mathcal{F}) = \text{HV}_{\boldsymbol{r}}(\mathcal{F})/\prod_{i=1}^{M} |r_i - z_i|$, where $\boldsymbol{z}$ is an ideal point satisfying $z_i < \min\{f_i(\boldsymbol{x})|\boldsymbol{f}(\boldsymbol{x}) \in \mathcal{F}\}$ (or $z_i > \max\{f_i(\boldsymbol{x})|\boldsymbol{f}(\boldsymbol{x}) \in \mathcal{F}\}$ for the maximization problem), $\forall i \in \{1, \ldots, M\}$. All methods share the same $\boldsymbol{r}$ and $\boldsymbol{z}$ for a MOCO problem, as given in Table 8.

## F  INSTANCE AUGMENTATION

In the inference phase, instance augmentation (Lin et al., 2022a) can be applied to boost the performance. Specifically, an instance can be transformed into others that share the same optimal solution. All transformed instances are then solved and the best solution is finally selected. An instance of Bi-CVRP has 8 transformations with respect to the 2-dimensional coordinates, i.e., $\{(x, y), (y, x), (x, 1-y), (y, 1-x), (1-x, y), (1-y, x), (1-x, 1-y), (1-y, 1-x)\}$. An instance of $M$-objective TSP has $8^M$ transformations due to the full permutation of $M$ groups of 2-dimensional coordinates.

## G  DETAILED RESULTS ON BENCHMARK INSTANCES

The detailed results of our method and other baselines on benchmark instances are provided in Table 9, which reveal the superior generalization capability of our method.

Table 9: Detailed results on benchmark instances.

| Method | KroAB100 | | | KroAB150 | | | KroAB200 | | |
|---|---|---|---|---|---|---|---|---|---|
| | HV↑ | Gap↓ | Time↓ | HV↑ | Gap↓ | Time↓ | HV↑ | Gap↓ | Time↓ |
| WS-LKH | **0.7022** | **-0.36%** | 2.3m | **0.7017** | **-1.05%** | 4.0m | **0.7430** | **-2.07%** | 5.6m |
| MOEA/D | 0.6836 | 2.30% | 5.8m | 0.6710 | 3.37% | 7.1m | 0.7106 | 2.38% | 7.3m |
| NSGA-II | 0.6676 | 4.59% | 7.0m | 0.6552 | 5.65% | 7.9m | 0.7011 | 3.68% | 8.4m |
| MOGLS | 0.6817 | 2.57% | 52m | 0.6671 | 3.93% | 1.3h | 0.7083 | 2.69% | 1.6h |
| PPLS/D-C | 0.6785 | 3.03% | 38m | 0.6659 | 4.10% | 1.4h | 0.7100 | 2.46% | 3.8h |
| MORAM | 0.6723 | 3.92% | 2s | 0.6603 | 4.91% | 2s | 0.6955 | 4.45% | 2s |
| DRL-MOA | 0.6903 | 1.34% | 10s | 0.6794 | 2.16% | 18s | 0.7185 | 1.29% | 23s |
| MDRL | 0.6881 | 1.66% | 10s | 0.6831 | 1.63% | 17s | 0.7209 | 0.96% | 23s |
| EMNH | 0.6900 | 1.39% | 9s | 0.6832 | 1.61% | 16s | 0.7217 | 0.85% | 23s |
| PMOCO | 0.6878 | 1.70% | 9s | 0.6819 | 1.80% | 17s | 0.7193 | 1.18% | 23s |
| WE-CA | 0.6959 | 0.54% | 9s | 0.6898 | 0.66% | 19s | 0.7232 | 0.65% | 25s |
| CNH | 0.6947 | 0.71% | 25s | 0.6892 | 0.75% | 35s | 0.7250 | 0.40% | 41s |
| WE-CA-U | 0.6948 | 0.70% | 9s | 0.6924 | 0.29% | 19s | 0.7317 | -0.52% | 26s |
| MDRL-Aug | 0.6950 | 0.67% | 13s | 0.6890 | 0.78% | 19s | 0.7261 | 0.25% | 28s |
| EMNH-Aug | 0.6958 | 0.56% | 12s | 0.6892 | 0.75% | 18s | 0.7270 | 0.12% | 27s |
| PMOCO-Aug | 0.6937 | 0.86% | 12s | 0.6886 | 0.84% | 19s | 0.7251 | 0.38% | 32s |
| WE-CA-Aug | 0.6997 | 0.00% | 14s | 0.6944 | 0.00% | 23s | 0.7279 | 0.00% | 38s |
| CNH-Aug | 0.6980 | 0.24% | 30s | 0.6938 | 0.09% | 37s | 0.7303 | -0.33% | 54s |
| WE-CA-U-Aug | 0.6990 | 0.10% | 14s | 0.6957 | -0.19% | 23s | 0.7349 | -0.96% | 39s |

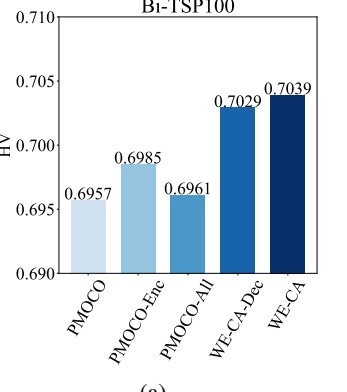

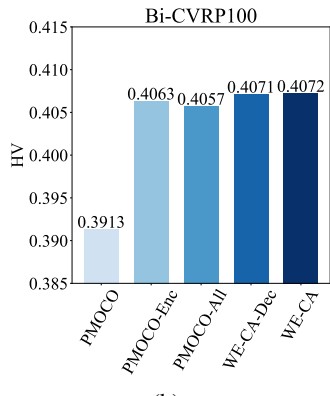

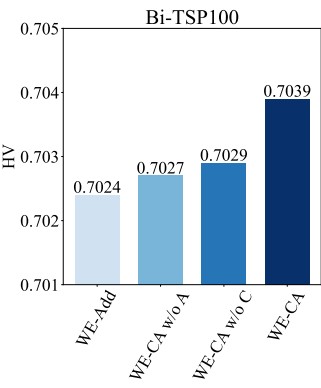

Figure 5: Effect of different weight embedding models.

(a)

(b)

Figure 4: Effect of weight-instance interaction via different parts of the model. (a) Bi-TSP100. (b) Bi-CVRP100.

# H DETAILED RESULTS OF DIFFERENT PATTERNS OF WEIGHT-INSTANCE INTERACTION

The results of weight-instance interaction via different parts of the model are presented in Figure 4. WE-CA outperforms WE-CA-Dec, which only employs the decoder to interact weight information with instance information. In addition, PMOCO-Enc and PMOCO-All, both using the whole model for weight-instance interaction, are both superior to PMOCO, but inferior to WE-CA. These results illustrate that our weight embedding method using the the whole model is an effective way for weight-instance interaction. Moreover, as shown in Figure 5, WE-CA surpasses the basic addition model (WE-Add), conditional embedding model (WE-CA w/o A), and attention model (WE-CA w/o C), which verifies our model design.

Table 10: Results of different numbers of weight vectors during inference.

| Method | Tri-TSP20 | | | Tri-TSP50 | | | Tri-TSP100 | | |
|---|---|---|---|---|---|---|---|---|---|
| | HV↑ | Gap↓ | Time↓ | HV↑ | Gap↓ | Time↓ | HV↑ | Gap↓ | Time↓ |
| WS-LKH | 0.4712 | 0.00% | 12m | 0.4440 | -0.20% | 1.9h | 0.5076 | -0.79% | 6.6h |
| PMOCO ($N = 105$) | 0.4693 | 0.40% | 5s | 0.4315 | 2.62% | 8s | 0.4858 | 3.53% | 18s |
| WE-CA ($N = 105$) | 0.4707 | 0.11% | 5s | 0.4389 | 0.95% | 9s | 0.4976 | 1.19% | 19s |
| PMOCO ($N = 1035$) | 0.4735 | -0.49% | 34s | 0.4460 | -0.65% | 1.1m | 0.5052 | -0.32% | 2.9m |
| WE-CA ($N = 1035$) | 0.4745 | -0.70% | 36s | 0.4531 | -2.26% | 1.2m | 0.5171 | -2.68% | 3.0m |
| PMOCO ($N = 10011$) | 0.4744 | -0.68% | 4.9m | 0.4497 | -1.49% | 10m | 0.5110 | -1.47% | 27m |
| WE-CA ($N = 10011$) | **0.4753** | **-0.87%** | 5.4m | **0.4566** | **-3.05%** | 11m | **0.5225** | **-3.75%** | 29m |

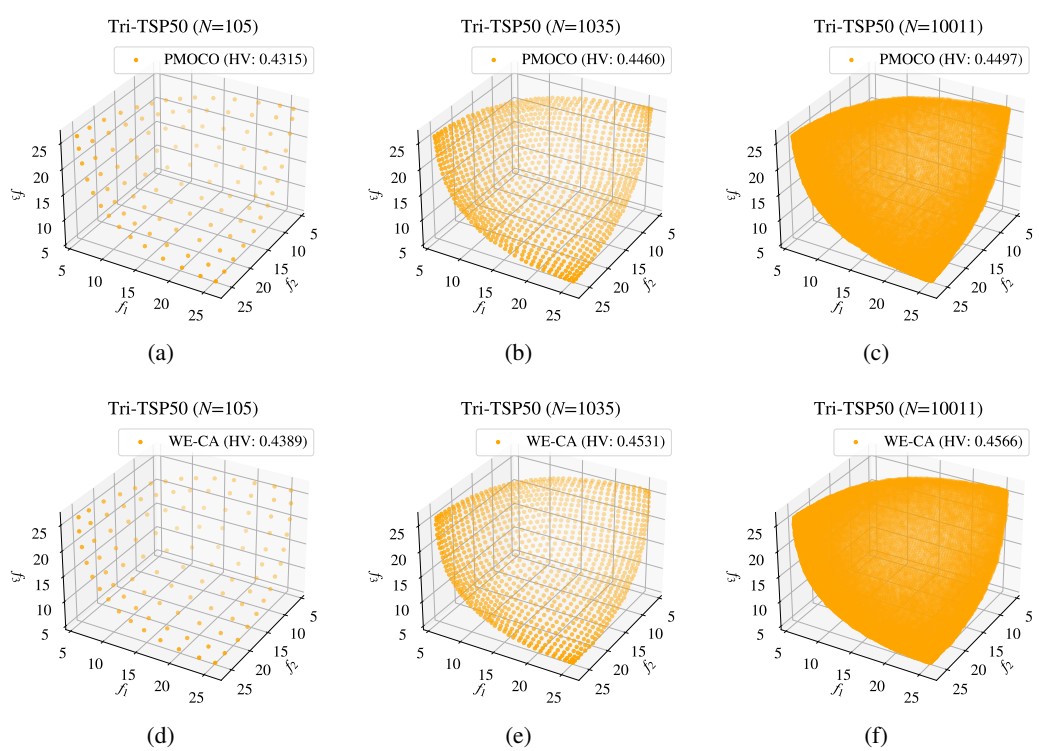

Figure 6: Pareto fronts derived by different numbers of weight vectors on Tri-TSP50. (a) PMOCO ($N = 105$). (b) PMOCO ($N = 1035$). (c) PMOCO ($N = 10011$). (d) WE-CA ($N = 105$). (e) WE-CA ($N = 1035$). (f) WE-CA ($N = 10011$).

# I HYPERPARAMETER STUDY

## I.1 STUDY ON THE NUMBER OF WEIGHT VECTORS DURING INFERENCE

Our method has high flexibility to deal with impromptu weight vectors during inference. We study the effect of the number of weight vectors that are all generated by the Das & Dennis (1998) method. According to Table 10 and Figure 6, our method can produce a denser Pareto front with better performance when using more weight vectors. Commendably, due to the smaller optimality gaps of our method on subproblems, our WE-CA with 1035 weight vectors even outperforms PMOCO with 10011 weight vectors, although they can both deliver well-distributed solutions.

To visually demonstrate the optimality of our WE-CA, we project the Pareto fronts at 2-dimensional planes, i.e., the $f_1$–$f_2$, $f_1$–$f_3$, and $f_2$–$f_3$ plane, as depicted in Figure 7. Apparently, all three objective values of the solutions delivered by WE-CA are smaller than that by PMOCO, which again verifies the superiority of our WE-CA for solving subproblems.

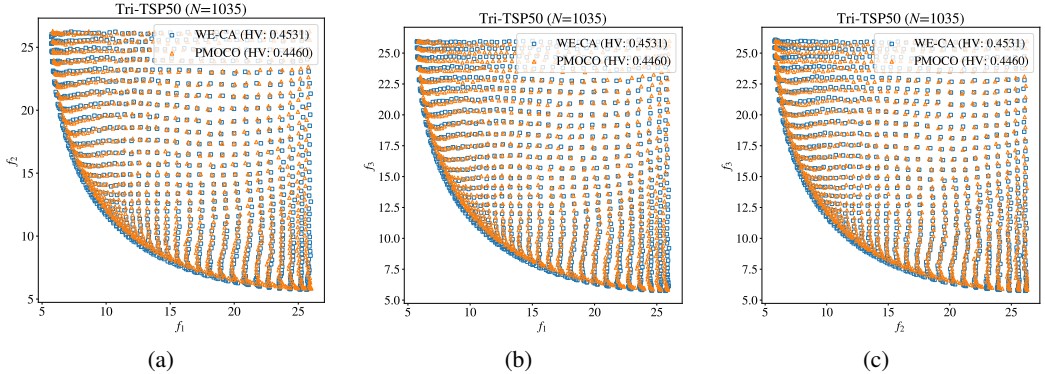

Figure 7: Pareto fronts obtained by WE-CA and PMOCO on Tri-TSP50. (a) Projection at the $f_1$–$f_2$ plane. (b) Projection at the $f_1$–$f_3$ plane. (c) Projection at the $f_2$–$f_3$ plane.

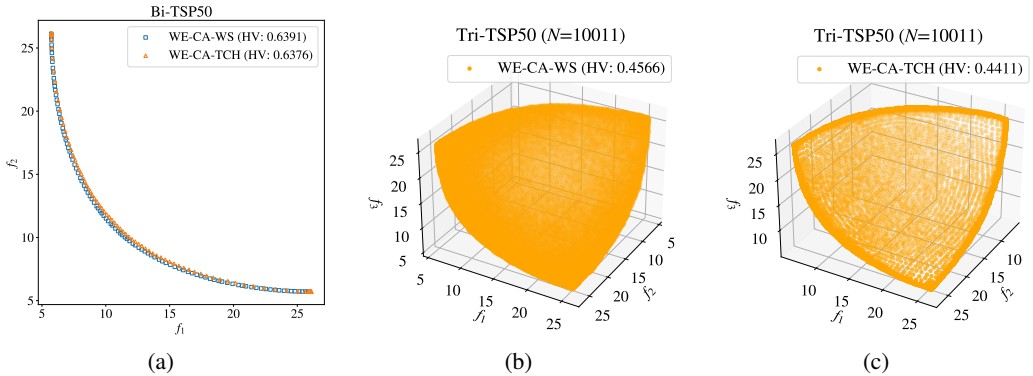

Figure 8: Results of different scalarization methods. (a) Pareto fronts on Bi-TSP50. (b) Pareto fronts obtained by WE-CA-WS on Tri-TSP50. (c) Pareto fronts obtained by WE-CA-TCH on Tri-TSP50.

## I.2  STUDY ON THE SCALARIZATION METHOD

Our decomposition-based method can adopt various scalarization methods, including the representative weighted sum (WS) and Tchebycheff (TCH). WS uses the simplest linear combination to effectively handle convex Pareto fronts, as follows,

$$\min_{\boldsymbol{x}\in\mathcal{X}} g_{\text{ws}}(\boldsymbol{x}|\boldsymbol{\lambda}) = \sum_{m=1}^{M} \lambda_m f_m(\boldsymbol{x}). \tag{13}$$

By contrast, TCH can tackle the concave $\mathcal{PF}$, but it would lead to a more complicated scalarized objective, as follows,

$$\min_{\boldsymbol{x}\in\mathcal{X}} g_{\text{tch}}(\boldsymbol{x}|\boldsymbol{\lambda}) = \max_{1\leq m\leq M} \{\lambda_m|f_m(\boldsymbol{x}) - z_m|\}, \tag{14}$$

where $\boldsymbol{z}$ is an ideal point satisfying $z_m < \min_{\boldsymbol{x}\in\mathcal{X}} f_m(\boldsymbol{x})$.

The results in Figure 8 indicate that WS is a simple yet superior scalarization method to TCH for the studied problems. Although TCH has stronger theoretical properties, the raised complexity of the scalarized subproblems leads to more difficulty in finding the optimal solution, as shown in Figure 8(a). In addition, Different scalarization methods result in different solution distributions. As demostrated in Figures 8(b) and 8(c), WS generates more uniformly distributed solutions, while TCH produces solutions that are sparser internally and denser at the edges.

