# OpenReview forum: "Rethinking Neural Multi-Objective Combinatorial Optimization via Neat Weight Embedding"
_ICLR.cc/2025/Conference — ICLR 2025 Poster_

### Official Review · Reviewer_pMit · 2024-10-31

**Soundness:** 3
**Presentation:** 4
**Contribution:** 4
**Rating:** 8
**Confidence:** 4

**Summary:**

The paper presents a novel weight embedding approach for neural multi-objective combinatorial optimization (MOCO) to address the optimality challenges observed in current decomposition-based methods. It focuses on capturing weight-instance interaction through a weight-specific representation learned directly within the neural model. Two model variations—Weight Embedding with Addition (WE-Add) and Weight Embedding with Conditional Attention (WE-CA)—are introduced. These models simplify MOCO by avoiding complex auxiliary techniques and showcase state-of-the-art performance across several MOCO problems, specifically the multi-objective traveling salesman, capacitated vehicle routing, and knapsack problems.

**Strengths:**

- Direct weight embedding is a fresh perspective in MOCO, addressing a gap in existing neural approaches that often require complex multi-model techniques.
- The paper provides thorough experimental results, comparing its models with state-of-the-art baselines (including multi-model, single-model, and heuristic-based methods) on three classic MOCO problems across different scales.
- The proposed WE-CA model achieves superior performance in terms of hypervolume (HV) and execution time, particularly highlighting its generalization capabilities across problem sizes.
- The models eliminate the need for size-aware embedding mechanisms, thus simplifying the optimization process.

**Weaknesses:**

- Real-world applications with complex constraints are acknowledged as challenging for this approach. Further exploration into handling such constraints would enhance the paper's practical relevance.

- The paper’s unified training model, WE-CA-U, provides promising generalization across problem sizes, but more discussion on its failure cases (where applicable) would improve understanding of its limitations.

**Questions:**

1-  In cases where the unified model fails to generalize effectively to certain sizes, could you provide more details on why this happens? Are there specific problem characteristics or settings that lead to these limitations?

2- Could you elaborate on how the conditional attention layer facilitates weight-instance interaction? While the paper describes the feature-wise linear projection mechanism, more insight into its layer-wise influence on embeddings would clarify how it improves optimality across subproblems.

---

### Official Review · Reviewer_CdjW · 2024-11-03

**Soundness:** 3
**Presentation:** 2
**Contribution:** 3
**Rating:** 6
**Confidence:** 2

**Summary:**

The paper presents a method for solving Neural Multi-Objective Combinatorial Optimization (MOCO) using a "neat" weight embedding approach. The authors argue that existing MOCO models are limited in their ability to effectively optimize weight-specific subproblems due to complex learning techniques and significant optimality gaps. Their proposed method learns weight-specific representations through a simpler weight embedding technique, capturing weight-instance interactions. Two models instantiate this approach: one with addition-based weight embedding and another with conditional attention. Experimental results demonstrate the method’s performance on benchmark MOCO problems, showing significant improvements in generalization across different problem sizes.

**Strengths:**

This work has extensive experiments that show the effectiveness of the proposed method and show strong cross-size generalization capabilities.

**Weaknesses:**

1. The authors could a clear definition of what they mean by "neat" in the context of their work, and highlight specific sections where they could elaborate on how their method contrasts with the complexity of existing approaches.
2. In the Methods section, while the structure of the proposed architecture is described, I would like to see a more detailed explanation of why each component is expected to improve performance. Specifically, theoretical justifications for key components such as the addition-based weight embedding and the conditional attention mechanism would be helpful. Some discussion of how the proposed components address specific limitations of previous approaches would also strengthen the work.
3. It would be valuable to see more ablation studies that isolate the contributions of each architectural component. For example, a fair experiment design that solely isolates the effect of the addition-based weight embedding.

**Questions:**

1. In Fig. 1, what is the input to get $h_c$ when $t=1$ in the decoder?
2. In Table 2, the proposed method appears to be slower than PMOCO. Does this indicate that the efficiency is worse compared to the baseline (given the 'neatness')?

---

> ### Comment · Reviewer_CdjW · 2024-11-26
>
> Thank you for providing detailed responses to my questions. My original concerns have been addressed. I am pleased to increase my score in favor of acceptance.

---

### Official Review · Reviewer_owYn · 2024-11-03

**Soundness:** 3
**Presentation:** 3
**Contribution:** 3
**Rating:** 6
**Confidence:** 3

**Summary:**

This paper introduced a new way for directly learning weight-specific representations, thereby improving the handling of decomposed subproblems. The authors designed two models for weight embedding: one is an additive embedding model, which performs embedding through simple addition operations; the other is a conditional attention model, which more accurately captures the interaction between weights and instance information through a conditional attention mechanism.

**Strengths:**

The weight embedding method proposed in this paper directly learns weight-specific representations, avoiding tedious adjustments and high computational costs, while improving performance without increasing model complexity.

The weight embedding method not only performs well across various problem scales but also shows strong generalization across different scales (such as varying numbers of nodes or task complexity). This capability allows the model to maintain good optimization performance when encountering problems of different scales or new challenges, demonstrating high adaptability. The additive weight embedding and conditional attention weight embedding models designed in the paper are not only straightforward but also adaptable to various MOCO tasks.

The authors also provides a lot of experiments for validation, showing the superority for their performance.

**Weaknesses:**

I think it would be beneficial to include more theoretical discussions. For example, the paper mentions that the weighted approach can improve generalization; adding a proof for the generalization bound would make the results more convincing. Additionally, when the number of classes approaches infinity, will this weighting approach converge to the average weight?

**Questions:**

For larger-scale problems, such as those with a large number of objective functions or high dimensions, how efficient is weight embedding? If the number of variables is quite large, could this impact the precision of weight learning? Will it still lead to improvements in training results?

Do the authors plan to release the code for validation? I believe that such a detailed comparison could be a great contribution to the community.

---

### Official Review · Reviewer_uh4f · 2024-11-04

**Soundness:** 3
**Presentation:** 3
**Contribution:** 3
**Rating:** 8
**Confidence:** 3

**Summary:**

The paper introduces a novel but simple neural multi-objective combinatorial optimization (MOCO) method. Specifically, the paper proposes a single-model method which can effectively solve MOCO problems (such as the multi-objective variants of the Traveling Salesman Problem or Capacitated Vehicle Routing Problem). This model is capable of learning the interaction of the problem instances with the weight vectors that are provided to decompose the problem into smaller, scalarized subproblems. At inference time, this allows the user to specify N weight vectors along with the problem instance, thereby producing a Pareto front of solutions. The authors introduce two variations of their method. In the first approach, named WE-Add, the interaction of the weight vectors and the node features is captured by simply adding their linear projections to get the node embeddings in the encoder of the model. In the second approach, WE-CA, the authors leverage a conditional attention model to capture the interaction of the instance and the weight vector. First, node embeddings conditioned on the weight vectors are derived through feature-wise affine transformations of the linear projections of the node features and weight vectors. Then, these embeddings are passed through standard transformer encoder layers, with multi-headed attention, instance normalization and feed forward networks. The authors demonstrate that this model not only reduces the optimality gaps of the subproblems but can also generalize well to problems of different sizes.

**Strengths:**

### Originality
The method of deploying "conditional attention" as proposed in the paper is simple and novel.

### Quality
With the exception of the points discussed in the Weaknesses Section, the paper is of good quality.
1. The paper features a comprehensive list of experiments. It discusses variations of several important problems, such as 20, 50, and 100 node variants of the bi- and tri-objective Traveling Salesman Problem (Bi-TSP and Tri-TSP), bi-objective Capacitated Vehicular Routing Problem (Bi-CVRP), and bi-objective Knapsack Problem (Bi-KP). The paper also demonstrates the out-of-distribution generalization for 150 and 200 node variants of Bi-TSP.
2. The authors justify their method which uses *conditional attention* by running ablation studies for its important components, such as *conditional embeddings* and *attention*. The experiments show that the combination of both these ideas work better than either one in isolation.


### Clarity

The paper is well-written. The ideas are communicated clearly. For example, Section 4.1 explains the base model that is used, and then builds on it in Section 4.2 to explain the model with conditional attention, making it easy to follow.

### Significance
The contributions of the paper are significant:
1. The simplicity of the method is commendable.
2. The proposed method shows strong performance compared to the baselines, showing smaller optimality gaps for the subproblems and higher hypervolumes, with comparable or faster solving times.
3. Also interesting is the finding that a unified model trained this way generally performs better than models trained for problems of specific sizes.

**Weaknesses:**

The choice of reference points for evaluating the authors' methods and the baselines raises a concern. In the paper, it is mentioned that CNH [1] bears some similarities to the authors' approach. The CNH paper also evaluates the bi- and tri-objective Traveling Salesman Problem as well as the bi-objective Capacitated Vehicle Routing Problem. Additionally, six out of the twelve baseline methods listed in Table 2 of this paper are also present in Table II of the CNH paper: MOED/D, NSGA-II, MOGLS, DRL-MOA, PMOCO, and PMOCO-Aug. However, the reference points for calculating hypervolume (HV) in this paper differ from those in the CNH paper, making direct comparisons with their results impossible. For reference, please see the table below.

| Problem   | Size | Reference Point (this paper)          	| Reference Point (CNH paper)  	|
|-----------|------|----------------|----------------------|
| Bi-TSP	| 20   | (20, 20)   	| (15, 15)        	|
|       	| 50   | (35, 35)   	| (30, 30)        	|
|       	| 100  | (65, 65)   	| (60, 60)        	|
|       	| 150  | (85, 85)   	| (90, 90)                	|
|       	| 200  | (115, 115) 	| (120, 120)                	|
|       	|  	|            	|                  	|
| Bi-CVRP   | 20   | (30, 4)    	| (15, 3)         	|
|       	| 50   | (45, 4)    	| (40, 3)         	|
|       	| 100  | (80, 4)    	| (60, 3)         	|
|       	|  	|            	|                  	|
| Tri-TSP   | 20   | (20, 20, 20)   | (15, 15, 15)    	|
|       	| 50   | (35, 35, 35)   | (30, 30, 30)    	|
|       	| 100  | (65, 65, 65)   | (60, 60, 60)    	|

Employing the same reference points as the CNH paper would enable a direct comparison and lend additional credence to the findings if they align with the established results.  Without this alignment, and in the absence of publicly available code, it is difficult to verify the results. Addressing this issue would greatly enhance the rigor and transparency of the paper.

I would be glad to reconsider my review if the authors could either provide results using the same reference points as the CNH paper or offer a clear justification for the reference points chosen in this study. This is the concern that has informed my rating for Soundness.

[1] Mingfeng Fan, Yaoxin Wu, Zhiguang Cao, Wen Song, Guillaume Sartoretti, Huan Liu, and Guohua Wu. Conditional neural heuristic for multiobjective vehicle routing problems. IEEE Transactions on Neural Networks and Learning Systems, 2024.

Other concerns about the paper are thus:
1. Figure 2 is intended to show the Pareto fronts, but it doesn’t. For instance, in the figure for KroAB200 (right), the results for CNH-Aug, represented by green circles, show that the left-most point appears to Pareto dominate all other CNH-Aug points. There are more examples across the three figures.
2. Section 3.1 and parts of 3.2 bear significant resemblance to the Section 3.1 and 3.2 of the paper on PMOCO [2]. However, I have overlooked this as they simply discuss the problem formulation.
3. The authors' use of subjective language, such as "the weight embedding are ingeniously incorporated into node embeddings" and referring to alternative approaches as "clumsy multi-model methods" detracts from the objectivity of the paper. A more impartial tone that directly outlines the methods would enhance the clarity and professionalism of the writing.

[2] Xi Lin, Zhiyuan Yang, and Qingfu Zhang. Pareto set learning for neural multi-objective combinatorial optimization. In International Conference on Learning Representations, 2022a.

A few other suggestions to improve the clarity and ease of reading the paper:
1. It would benefit the reader if it is explained that the solution, represented as a sequence $\( \pi = \{ \pi_1, \dots, \pi_T \} \)$, is simply a permutation of the nodes for the Traveling Salesman Problem.
2. $P(\pi | \lambda, s)$ is introduced in Section 3.2, but $s$ is only described in 4.1.
3. “IN” (instance normalization) used in Eq (2) is never explicitly stated anywhere.
4. Discussing the venue of publication in Table 1 is not necessary, in my opinion. Neither is the description of baseline methods as "complex" and the authors' method as "neat". Parameters is misspelled in the column header.

**Questions:**

Can the authors justify the reference points that were chosen for the experiments?

---

### Meta-Review · Area_Chair_pZRc · 2024-12-17

**Metareview:**

The paper proposes a novel weight embedding method for neural multi-objective combinatorial optimization. It addresses the suboptimal performance of existing decomposition-based methods in weight-specific subproblems. By introducing two models, WE-Add and WE-CA, the authors effectively capture weight-instance interactions. Experimental results on multiple MOCO problems show that their approach achieves better generalization and performance compared to state-of-the-art baselines.

Strengths of the paper include its originality in the weight embedding concept, comprehensive experimental validation, and clear communication of ideas. However, it has some areas for improvement. Theoretical justifications could be enhanced, and certain terms and design choices need more detailed explanations. Overall, the paper's contributions outweigh its weaknesses, and with the authors' satisfactory responses to reviewer concerns, it is worthy of acceptance.

**Additional Comments On Reviewer Discussion:**

During the discussion, reviewers raised several key points. Concerns were expressed about the clarity of the paper, such as the definition of terms and the explanation of model components. Questions were also asked regarding the theoretical basis and the efficiency of the method in different scenarios. The authors addressed these concerns by providing more detailed explanations, conducting additional experiments, and justifying their design choices. They also made efforts to improve the clarity of the writing and uploaded the source code for reproducibility. Considering these responses, it was evident that the authors were committed to enhancing the quality of the paper, which contributed to the decision to accept it.

---

### Decision · Program_Chairs · 2025-01-22

Accept (Poster)